# Preference for Masculine or Feminine Gender Roles and Its Relationship to Well-Being in Transgender Persons: Comparing Pre-Treatment, Hormonal Therapy, and Post-Surgery Groups

**DOI:** 10.3390/bs10060100

**Published:** 2020-06-11

**Authors:** Sergey Kumchenko, Elena Rasskazova, Aleksander Tkhostov, Vadim Emelin

**Affiliations:** Faculty of Psychology, Lomonosov Moscow State University, Moscow 125009, Russia; e.i.rasskazova@gmail.com (E.R.); tkhostov@gmail.com (A.T.); emelin@mail.ru (V.E.)

**Keywords:** transgender persons, stages of the transgender transition, well-being, gender roles

## Abstract

This article aims at identifying different preferences for gender roles in transgender persons and the associations of those preferences with well-being at the different stages of medical transition. A total of 148 Russian transgender subjects (64 in pre-treatment, 41 in hormonal therapy, and 43 in hormonal therapy after some surgery) and 120 Russian cisgender persons filled out the Minnesota Multiphasic Personality Inventory-2. The transgender participants were less satisfied with their lives than the cisgender individuals, and less frequently preferred masculine gender roles, which were related to higher well-being in both groups. However, both preference for masculine gender roles and well-being were higher in the hormonal therapy group, and especially after surgery, regardless of whether they were females transitioning into males or vice versa. In the transwomen, having only hormonal therapy was associated with the poorest satisfaction, while those undergoing hormonal therapy after surgery were much more satisfied with their lives. Those differences are reasonable, considering the visible changes in the body and mood after hormonal therapy in transmen, and because results after surgery are more promising for transwomen than for transmen.

## 1. Introduction

Gender dysphoria is traditionally defined as a stable dysphoric feeling of a mismatch between one’s sex assigned at birth and gender self-determination, as well as a constant desire to be of the opposite sex and to change oneself by medical or surgical intervention. Due to the severe stigmatization related to gender dysphoria [1,2], this condition is described today as “transgender” to include any stable experiences of gender identity typical of the opposite sex, without any reference to pathology. Empirical studies indicate the importance of taking into account the degree of desired changes: Male to female or vice versa [3].

Transgender people frequently experience social and family disapproval [4,5], report poor well-being [6,7], and are at higher risk of suicide. This is especially pronounced in Russia, where negative attitudes toward gender transition are widespread and are caused by the lack of knowledge in the field. For example, this is evident in the employment of transgender people in Russia. Sociological surveys among transgender individuals indicate that “many respondents have no opportunity to get a well-paid job in accordance with their specialty due to a negative attitude toward their transgender nature on the part of employers” [8] (p. 31). As a result, transgender persons face misunderstanding on the part of their friends and relatives and avoid discussing their problems even with clinicians. Due to the fact that their financial situation does not allow renting apartments on their own, many transgender people have to live with their parents and/or relatives, who have a negative attitude toward the transgender phenomenon; they have to hide their transgender identity from their parents and/or relatives [8] (p. 31). Transphobia and discrimination in medical institutions lead to the fact that transgender patients prefer not to have any contact with doctors at all [9] (p. 30). It has been demonstrated that hormonal therapy and surgery lead to better well-being in transgender people [10,11], although the general level of their well-being has still been lower than that in the population as a whole [12,13], and there are some studies with contradictory results [14,15]. In our view, further investigations of well-being in transgender people require larger samples, taking into account preferences in gender roles, the degree of desired change, and the stage of medical transition (pre-medication group, hormonal therapy group, and people after some surgery).

The present study aims to identify differences in gender roles in transgender people and their relationship with well-being at the different stages of the medical transition. We hypothesize that:Russian people who define themselves as transgender prefer roles typical of the desired gender and reject roles typical of their sex assigned at birth.The lower level of well-being in Russian transgender people in comparison with Russian cisgender persons is more pronounced in the pre-treatment group and the least pronounced in the after-surgery group.The lower level of well-being in Russian transgender people is not fully explained by their preferences for masculine or feminine gender roles.

Finally, we hypothesize that hormonal therapy rapidly leads to the perception of the desired changes in transmen, while gender affirmation surgery is more effective for transwomen. We suppose that their well-being is at its peak during the hormonal therapy in transmen and after surgery in transwomen.

## 2. Methods

The survey participants included 148 transgender subjects (46 assigned male at birth, or transwomen, and 102 assigned female at birth, or transmen) aged 14–56 years (mean age: 23.93 ± 7.85 years); 120 cisgender persons (40 assigned males at birth, or cismen, and 80 assigned females at birth, or ciswomen) aged 14–56 years (mean age: 26.11 ± 7.62 years) participated as well. Thus, the whole sample totaled 268 subjects.

The pre-treatment group comprised 64 transgender persons (12 transwomen and 52 transmen) with no experience of hormonal therapy or surgery; 41 transgender subjects (16 transwomen and 25 transmen) were in the hormonal therapy group; and 43 individuals (18 transwomen and 25 transmen) were in the group that had had hormonal therapy and some surgery for a gender transition. The recruitment method was that transgender persons contacted other transgender people via Internet communities. These were participants of closed trans-communities in social networks (Facebook, Vkontakte.ru, Transgender.ru, etc.) and in Vk.com, mainly: “Transgender secrets”, “FTM”, “T-thoughts”, etc. Another part of the respondent group was found offline in the Moscow Non-Commercial Foundation “Transgender”. The participants were invited for an interview with a clinical psychologist. A self-definition as transgender and a strong feeling of gender dysphoria in the present or the past were criteria for inclusion. The exclusion criteria were ordinary doubts about gender, diagnosed severe mental illness (except for reactions to acute stress, or affective and personality disorders), and any possible severe mental disorder diagnosed during the interview with the clinician from the research group. The cisgender group comprised people whose gender identity matched the sex assigned at birth.

All subjects gave their informed consent for inclusion before they participated in the study. The study was conducted in accordance with the Declaration of Helsinki, and the protocol was approved by the Ethics Committee of the Lomonosov Moscow State University Faculty of Psychology on 15 April 2019 (project identification code ECLMSUFP-15-03-19); it met the requirements of the Code of Ethics of the Russian Psychological Society.

The participants filled in the Gender Roles—Male and Female scales of the Minnesota Multiphasic Personality Inventory-2, measuring preferences for traditionally masculine and feminine gender roles [16]. Well-being was assessed in accordance with E. Diener’s model [17], including a cognitive component of satisfaction with life [18] and the emotional components of positive and negative emotions [19].

The data were processed in the SPSS Statistics 23.0 program. A 2 (sex assigned at birth) × 2 (group: Transgender or cisgender) analysis of variance (ANOVA) was used to reveal gender- and transgender-related differences in preferences for gender roles and well-being. Correlations between well-being and preferences for gender roles were calculated separately for each group and compared by Pearson’s Chi-square test. A 2 (sex assigned at birth) × 3 (stage of medical transition: Without hormonal therapy, with hormonal therapy, or both with hormonal therapy and post-surgery) ANOVA was used to identify differences in well-being and gender roles in those in the hormonal therapy group and the after-surgery group. The significance level to reject null hypothesis was *p* < 0.05.

## 3. Results

### 3.1. Preferences for Gender Roles and the Association of the Preferences with Well-Being in Transgender and Cisgender Participants

There were no major effects of sex assigned at birth on the preferences for gender roles.

It was interesting that, unlike the cisgender subjects, the transwomen scored higher on the GF scale (Gender Role—Feminine) and lower on the GM scale (Gender Role—Masculine) than the transmen. As Figure 1 shows, using Russian population norms [20], the cisgender individuals were close to the country’s medium 50-UT scores, while the transwomen were much higher on the GF scale and lower on the GM scale. The opposite was true for the transmen.

In general, the transgender subjects were much less satisfied with their lives and reported more negative and less positive emotions than the cisgender persons (Table 1).

In the cisgender individuals, the preference for masculine gender roles was correlated with higher satisfaction and positive emotions and lower negative emotions (|r| = 0.25–0.31 for the males and |r| = 0.28–0.47 for the ciswomen), while the preference for feminine gender roles was not correlated with well-being (r = −0.05–0.10) except for a weak correlation with negative emotions in the cismen (r = −0.23). In the transgender subjects, the general pattern of correlations remained the same: The preference for masculine gender roles was associated with better well-being, both for the transwomen (|r| = 0.26–0.43) and the transmen (|r| = 0.31–0.53), while the preference for feminine gender roles was not correlated with well-being (r = −0.02–0.12).

### 3.2. Stage of Transgender Transformations, Gender Roles, and Well-Being

The major effect of gender in the transgender group, regardless of the stage of transformation, was that the transmen preferred masculine gender roles more frequently and feminine gender roles less frequently than the transwomen (F = 15.89, *p* < 0.01, η^2^ = 0.10 and F = 46.39, *p* < 0.01, η^2^ = 0.25, respectively).

The major effect of the stage of transformation was that the transgender subjects of both genders in the hormonal therapy group and especially the after-surgery group preferred more masculine gender roles; they were more satisfied and happier than those who were not in hormonal therapy. The post hoc Scheffé contrast detected that after surgery, the patients were significantly more satisfied with their lives than the individuals in hormonal therapy alone and those in pre-treatment (*p* < 0.05). Moreover, they experienced more positive and less negative emotions in comparison with those taking no hormones (*p* < 0.05). There were no post hoc group differences in the preference for masculine gender roles.

There was an interaction effect between the stage of transformation and the sex assigned at birth (F = 8.38, *p* < 0.01, η^2^ = 0.11) on satisfaction with life. Satisfaction was higher in the transmen in hormonal therapy, and satisfaction in those in hormonal therapy and after surgery was the highest in comparison with those who did not take any hormones or did not have surgery. Satisfaction in the transwomen was the highest in the group after surgery, but lowest in those in hormonal therapy alone. However, this interaction effect was not replicated for positive and negative emotions (Table 2).

## 4. Discussion

Although most of the transgender persons in our study had had no clinical diagnosis and no appointments with a psychiatrist, differences in their preference for gender roles speak in favor of their specific gender-related experience. The transwomen (male at birth) prefer feminine roles and reject masculine gender roles, and vice versa.

Comparison of the raw scores of the GF (Gender Role—Feminine) and GM (Gender Role—Masculine) scales indicates that transgender persons have more interests typical of the desired gender and fewer interests typical of their own sex at birth. This is consistent with the definition of the transgender phenomenon [3]. It is noteworthy that, on average, the preference for masculine gender roles in the transgender respondents is lower in comparison with the cisgender persons. That could partially explain their poor well-being, considering that those preferences are associated with higher satisfaction and more positive and less negative emotions. In fact, the data confirm that the transgender people need psychological support due to their poor well-being and negative experience. That is also proved by statistical surveys among transgender people in Russia [8,9].

The decision to use hormonal therapy and gender affirmation surgery is associated with both the preference for masculine gender roles and better well-being. That may be due to the specificity of Russian culture, which approves the manifestation of masculinity in both men and women. Perhaps for this reason, female-to-male (FtM) transgender persons significantly outnumber male-to-female (MtF) transgender individuals in Russia, unlike in the rest of the world [21]. This may point to a strong cultural influence on transgender people. The connection between the preference for masculine gender roles and high well-being may also be explained by the fact that to affirm a gender transition, the life of a transgender person requires courage, decisiveness, and initiative: “Identity is determined not by words but, largely, what a person is willing to sacrifice for it” [22]. It has been proven that men demonstrate the abovementioned characteristics more often. “Males are proud to do everything on their own” [23] (p. 241). Apparently, gender affirmation is a way of manifesting stereotypical masculine traits. Thus, we can hypothesize that the manifestation of only masculine features is a means to reduce the sense of uncertainty in Russian transgender people, since “gender dysphoria is a subtype of the sense of uncertainty, which can be overcome with help of a decisive attitude to their own life” [24] (p. 334). These are stereotypically considered as the display of masculinity. We also suppose that this conflicting perception of male gender roles is typical of transgender individuals in Russia, and is reduced and smoothed out by the end of the transgender transition.

By the end of the medical transition, the FtM transgender persons expressed their preference for masculine roles less than the cisgender men, and the MtF transgender participants accepted masculine traits. Probably, this is a matter of insufficient interiorization of gender roles, or it refers us to Sandra Bem’s concept of psychological androgyny: “In a society where rigid sex-role differentiation has already outlived its utility, perhaps, the androgynous person will come to define a more human standard of psychological health” [25] (p. 162). “Androgynous” individuals “might be both masculine and feminine depending on the situational appropriateness of these various behaviors” [25] (p. 155). Other authors also believe that psychological androgyny is a trait of transgender people [13]. We assume that gender preferences in transgender persons change during the medical transition. Before that transition, transgender individuals prefer strict adherence to gender stereotypes, beginning from the unambiguous opposite (of their own sex assigned at birth) and shifting toward an androgynous version of gender identity. This may reflect the dynamics of gender identity in terms of gender stereotypes. We may assume that gender dysphoria does not always reflect a desire to coincide with the chosen gender role, contrary to the generally accepted definition of the transgender phenomenon; sometimes, it is a particular form of psychological androgyny. Thus, the transgender phenomenon during the medical transition is not monolithic in its psychological properties. The following data speak in favor of that.

For the transwomen, hormonal therapy is associated with the poorest satisfaction with life. In our opinion, that is due to their strongly condemned androgynous appearance. In spite of the ongoing intake of hormones by transwomen, their appearance does not become feminine at the beginning, and that causes frustration. Reality does not match their expectations.

For the transmen, hormonal therapy leads to rapid desired changes in the body and mood, which is reasonably associated with a high increase in well-being. We hypothesize that there is an experience of control of their life and the joy of winning at one of the stages of transformation into a man. During our examination, the male sex hormones themselves had the effect of “a second puberty” in the form of disinhibition of the affective sphere.

However, further surgical intervention is not as promising for females as for males, and an understanding of its limitations might not be a positive experience. In the context of patriarchal Russian culture, FtM transgender persons are forced to feel their mismatch with the average man due to the lack of significant male anatomical features. Moreover, they experience frustration with the transgender path they have chosen, which they once overvalued. Reality does not match their expectations.

Perhaps some transgender persons overestimate the results of gender affirmation procedures, attributing mythological positive features to the transgender transition—for instance, patients in clinical institutions who feel they have increased personal significance [26].

In the case of the males (MtF), hormonal therapy has not made them more feminine, while their mood may become more negative because of the medication. This could explain why hormonal therapy may be a negative experience for transwomen in transition, while surgery is related to a high increase in well-being.

Thus, there is an ambiguous effect of the medical transition on transgender persons. In spite of the well-known advantage of hormone therapy for transgender individuals [3,10,11], we observe that this therapy is associated with poor well-being in some cases. Chinese colleagues [14,15] have recently come to similar conclusions [12,13,24]. Perhaps the issues here are cross-cultural differences in the transgender transition.

In general, there is a clinical and psychological reason to differentiate transgender individuals by the stages of their gender affirmation, since significant psychological differences are found between the stages of medical transition. We have also found these in other studies [27,28], taking into account that the gender affirmation stage is the basis for developing a psychotherapy program for transgender people. For instance, during the medical transition, transgender persons are more oriented toward solving particular problems of employment; an individual type of psychotherapy is optimum for them. Transgender individuals before the medical transition are more concentrated on the expression of their gender identity; group work is better for them [28]. The stage of gender affirmation implies different needs and different problems in transgender persons, requiring different approaches from experts.

The Modern Convention within the framework of depathologization of gender dysphoria [1] presents a clear view on this phenomenon as a variant of sexual health. Speaking about the variability of sexual health, we dare state that there are alternative variants of psychosexual ontogenesis. The transgender phenomenon is an alternative ontogeny that is characterized by its own periodization, and that includes the stages of transgender transition shaped by the different social situations of development described, in other contexts, by L. Vygotsky [29]. Consideration of the stages of transgender transition enables us to take into account the actual psychological needs of transgender patients and to predict further interaction during the development of complex rehabilitation and psychotherapy programs.

The limitations of our study are the employment of population norms in Russia without matching the participants with other demographic data, such as socioeconomic status and family support.

## 5. Conclusions

Transgender people prefer the roles typical of their affirmed gender and reject the roles related to the sex assigned at birth. In comparison with cisgender individuals, transgender persons are less satisfied with life and less frequently prefer the masculine gender roles associated with higher well-being in both groups. However, both the preference for masculine gender roles and well-being are higher in those in hormonal therapy, especially after surgery, regardless of whether they are females transitioning to males or vice versa. These data confirm that hormonal therapy in transmen could lead to a high increase in satisfaction with life, while the increase after surgery is minimal. In transwomen, hormonal therapy is correlated with the poorest satisfaction, while those after surgery are much more satisfied with their lives. These differences are reasonable, taking into account the visible changes in the body and mood after hormonal therapy in transmen and much more promising results after surgery for transwomen than for transmen.

## Figures and Tables

**Figure 1 behavsci-10-00100-f001:**
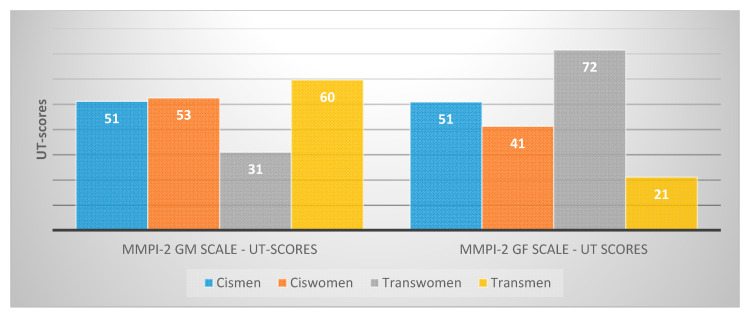
UT scores on Gender Role—Masculine (GM) and Gender Role—Feminine (GF) MMPI-2 (Minnesota Multiphasic Personality Inventory, 2nd version) scales for the cis- and transgender participants (according to Russian population norms).

**Table 1 behavsci-10-00100-t001:** Preferences for gender roles and well-being in the transgender and cisgender people.

Scales	Cisgender People				Transgender People				*Major Effect of Group ****		*Interaction between Group and Gender ****	
	Cisgender Men		Cisgender Women		Transwomen		Transmen					
	Mean	St. dev.	Mean	St. dev.	Mean	St. dev.	Mean	St. dev.	F	η^2^	F	η^2^
Satisfaction with life	21.28	5.06	21.83	5.82	16.52	7.11	16.12	5.72	45.38 **	0.15	0.37	0.00
Positive emotions	20.03	4.75	20.64	4.96	18.39	5.30	16.84	5.12	16.81 **	0.06	2.67	0.01
Negative emotions	15.33	4.05	15.73	5.15	17.65	5.33	19.38	5.28	2.10 **	0.07	0.99	0.00
MMPI-2 GM scale (raw scores)	35.08	5.19	27.60	6.28	26.26	8.12	30.61	6.76	1.97 **	0.04	45.74 **	0.15
MMPI-2 GF scale (raw scores)	25.10	3.54	31.18	4.88	31.30	5.19	25.00	5.06	0.00	0.00	95.19 **	0.26

** *p* < 0.01. *** A 2 (sex assigned at birth: Masculine versus feminine) × 2 (group: Transgender versus cisgender) analysis of variance (ANOVA) was used to reveal gender- and transgender-related differences in well-being and preferences for gender roles.

**Table 2 behavsci-10-00100-t002:** Well-being and preference for gender roles in the transgender participants at different stages of transformation.

Scales	Transgender People without Hormonal Therapy or Surgery				Transgender People in Hormonal Therapy				Transgender People in Hormonal Therapy and after Surgery				Major Effect of Stage of Transformation	
	Transwomen		Transmen		Transwomen		Transmen		Transwomen		Transmen			
	Mean	St. dev.	Mean	St. dev.	Mean	St. dev.	Mean	St. dev.	Mean	St. dev.	Mean	St. dev.	F	η^2^
Satisfaction with life	16.17	6.51	14.00	5.07	11.25	3.11	17.68	6.46	21.44	6.78	19.04	4.71	13.77 **	0.16
Positive emotions	17.08	6.08	15.54	5.17	17.19	4.75	17.04	5.25	20.33	4.89	19.16	4.30	5.17 **	0.07
Negative emotions	19.08	6.14	21.06	4.93	19.13	4.38	18.56	6.09	15.39	5.01	16.56	3.78	7.09 **	0.09
MMPI-2 GM scale (raw scores)	24.83	7.31	29.42	7.34	25.44	6.91	30.56	6.34	27.94	9.64	33.60	4.21	2.99 *	0.04
MMPI-2 GF scale (raw scores)	30.92	3.78	24.81	5.20	30.50	6.31	24.88	4.59	32.28	5.02	24.96	5.12	0.38	0.01

* *p* < 0.05, ** *p* < 0.01.

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
