# Peer review of "Preference for Masculine or Feminine Gender Roles and Its Relationship to Well-Being in Transgender Persons: Comparing Pre-Treatment, Hormonal Therapy, and Post-Surgery Groups"

_behavsci, 2020, doi:10.3390/bs10060100_

Round 1
Reviewer 1 Report
Thanks you very much for the opportunity to review The Preference of Masculine and Feminine Gender Roles... I enjoyed reading the paper and I think that the results are important and that the paper merits rapid publication. As an anthropologist, I particularly appreciate the space given to considering the effects of culture. I should note that since my own research is largely qualitative, I don't feel fully adequate to assess the statistical analysis presented.
One thing that the authors might want to note is that, at least in the US, the estimates for the number of transgender people is currently about 0.6%, so two to three orders of magnitude greater than what the authors present in line 31. The authors do note that much higher rates are reported but that's a huge difference.
Most of my problems with this paper are really about language. I've great respect for people publishing in a second language; something I've never had to do. But, the paper really needs to be carefully edited by a native English speaker. There are many small problems and, taken together they make it difficult to fully understand or assess the paper. Some fast examples, lines 43, 44; 67,68; 91; 120, 121; 127, 128; 186-190; and many others. I can provide editing details if the authors wish.
Some of the language difficulties may create issues that the authors do not intend. So for example, 162 talks about the "choice to be genderqueer." The use of "choice" in this context might be pretty contentious. Of course, every morning when I wake up, I have a choice of what I wear and how I perform my identity. However, performing identity is different than being. I'm not genderqueer, but if I was I doubt I'd consider it a choice. If the authors want to enter into the debate among those who claim such identifications are just aspects of individuals versus those that claim individuals choose them, OK, but I'd recommend against.
Or line 193 talks about "reprehensible androgynous physical appearance." I understand the authors to mean that androgynous physical appearance is negatively valued. However, reprehensible is a very strong word that usually implies individual moral failure. I don't think that's what the authors probably intend. Or line 199: I'm not sure the authors want to say "winning." Maybe "achieving." Winning implies a competition and, maybe it is. But, if so, it's something the authors might want to unpack.
So, for me, this is an important and interesting paper in need of close editing by a native speaker. I'd like to read a better-written version.
Author Response
Dear Reviewer,
Above of all, we would like to express your gratitude for your very kind attention, comments and remarks on your article about transgender people.
We tried to consider all your suggestions to fix it, eliminate all the faults and put in all the necessary quotations where it was possible:
* all the quotes are yellow;
* all the corrections are green;
* the passage about non-binary identities is transferred into the introduction;
* lines 39-42 include the updated data about the prevalence of transgender people:
* as for genderqueers, we would insist on the word "choice" because, we believe, gender identity is the result of choice and self-determination but not that of self-expression. This corresponds to the modern concept of social consruction of gender identity.
* 249-253 corrected the description of the look of MtF transgender people in the hormone therapy.
We hope that we have been able to make our article better.
Best regards,
All the authors

Reviewer 2 Report
This paper describes results of a survey regarding gender roles which was given to about 250 people, most of whom were transgender and compared the results of those with affirmed gender identity of a woman to those with the affirmed gender identity of a man, and also compared those who were transgender to those who were cisgender.
The paper provides some insight into well-being and gender roles, but also discusses other topics not directly related to the theme of the paper, which I think should be further evaluated for their need and relationship to the data presented.
The paper needs significant improvements in the language of it. There are multiple terms which are used incorrectly and could be offensive to some readers.
1- cisgender should not be used as a noun, only as an adjective. saying 'the cisgenders' is inaccurate, instead it should be phrased as 'cisgender people' - same with transgender
2- the wording of 'morphologically male' or 'initially male' is not commonly used in English or America when discussing people with male genitalia at birth. More common wording would be 'those assigned the sex male at birth.'
3- transexuality - this is listed as an offensive term and outdated term in multiple texts. transgender should be used instead.
4- sexual reassignment surgery is also outdated- current terminology is gender affirmation surgery
Title/abstract:
I think the title and the abstract should include that it was set in Russia since it is referenced in the paper specific aspects of Russian culture.
Introduction:
The first word of the introduction should be changed to 'gender dysphoria' to accurately reflect the DSM-V. In this paragraph - the authors reference the DSM-IV which is an outdated version since the DSM-V is available.
I also think the terminology should be presented more clearly and be used consistently throughout the manuscript. There are multiple words which are not defined and used interchangably. I recommend using the terminology of transwomen and transmen or Ftm and MtFTG. It is recommended to use the gender identity of TG people rather than their sex assigned at birth - repeatedly using morphologically male is counteracting this. Instead of referring to people who were assigned the sex male at birth and have a feminine gender identity as 'morphologically male', they should be described as 'transwoman'
The definition of non-binary and genderqueer should be moved from the discussion to the introduction and then clearly stated that this study excluded them in the methods. Especially since genderqueer has many meanings it should be defined as what it is meaning in the context of this paper.
Results:
Lines 96-101 are unclear.
Table 1: This table needs the headings reworded - cisgenders should be 'cisgender people', transgenders should be transgender people and the subcategories should be transmen and transwomen, and for consistency- the cisgender shoudl be sicgender women and cisgender men.
The titles of 'major effect of group' should have an asterics and explanation of what was compared and how in the table footer as well as the interaction between group and gender.
Additionally explanations of these scores including the range of the scale and what a high or lower number indicates.
Discussion:
-lines 176-192 are confusing and I'm not sure exactly what their purpose is- perhaps by defining genderqueer, gender non-binary they and more clearly explain psychological androgyny. The way I interpret it as written is that that gender non-binary is not a real concept and is currently a theory that some people may not identify as one of the binary genders.
Line 192-196 - citation needed
Line 197-201 - citations needed, specifically to support the last sentence in the paragraph.
Line 202-206 - citation needed
207-209 - seems like an extrapolation
Line 221-224 - cite
LIne 227-230 - explain how this would affect care - what should or do physcologists/psychiatrists do differently why is their stage of transition important - last sentence is vague - is there any evidence or studies showing this technique/practice is useful? or is this unsubstantiated theory?
-limitations about using population norms of Russia without matching participants or looking at other demographic data - like socioeconomic status, mental health, family support that factor into preferences

Author Response
Dear Reviewer,
Above of all, we would like to express your gratitude for your very kind attention, comments and remarks in relation to your article about transgender people.
We tried to consider all your suggestions to fix it, eliminate all the faults and put in all the necessary quotations where it was possible:
* all the quotes are in the yellow colour;
* all the corrections are green:
* the mentioned terms are changed;
* non-binary identities are transferred into the introduction;
* the results are edited;
* the text in the discussion which, in your opinion, required the third-party quotes, is changed to emphasize our authorship of the statements (Lines 249-270);
* Lines 232-236 comment on the concept of the psychological androgyny
* Lines 294-303-describes the practical application of the provision on the allocation of transgender people by stages of gender correction;
* limitations are mentioned.
We hope that we have been able to make our article better.
Best regards,
All the authors

Round 2
Reviewer 2 Report
Dear Authors,
While I do see improvements, there are still significant terminology concerns which I believe are very important to correct for this topic. While the clarity and accuracy is improved, there are still errors and uncited claims. I spent time reviewing the abstract and introduction and have attached my comments. I did not review the remainder of the article as from reviewing the introduction I have determined that it is not suitable for publication in it's current form.
I would consider identifying an English speaking person who works in this field and work closely with them and consider adding them as a co-author if you plan to publish your article in English.

Author Response
Dear reviewer,
We are grateful to you for your kind attention and comments to our paper. In reply to your letter we would like to inform you about the following:
* A native speaker re-corrected the entire text.
* Your comments on the terms were taken into account.
* The data about the prevalence were deleted as this was not essential for our work.
* The genderqueer identity is a choice as a result of social constructionism as opposed to biological determinants. We would insist on that fact that this is a choice of personality.
We hope that we have been able to make our article better.
Best regards,
All the authors

Round 3
Reviewer 2 Report
Hello,
Thank you for the incorporating improvements into the paper. The entire paper reads well and the ideas are easier to understand.
One major revision still remains - I think in the introduction, the term non-binary is still explained incorrectly. Non-binary does not relate to whether or not a person is taking hormones. Non-binary relates to their gender identity. Anyone who has a gender identity of man or woman (whether they are cis or trans) fits in one of the gender binaries. Anyone who identifies with a gender other than man or woman is considered gender non-binary. These genders can include genderqueer, agender, pangender, bigender, and many others.
Other than that, some additional comments are added for clarity. Thank you for your efforts in conducting this important research and your efforts of revisions to improve it.
Respectfully,

Author Response
Dear reviewer,
We are grateful to you for your kind attention and comments to our paper. In reply to your letter we would like to inform you about the following:
* all the corrections are yellow;
* the part about non-binary transgender people is removed due to the insignificance of this issue in our article;
* we did not perform any analysis before the study because there were no previous information on effect sizes we should expect in the Russian sample. So, we invited all the transgender persons who agreed to participate in it;
* we have stated in Methods what is considered to be statistically significant;
* we have removed those parts of the article with the insufficient significance;
* the Discussion uses the concept of psychological androgyny; this is not androgyny as gender but the psychological androgyny as the simultaneity of high indicators on the gender roles male and female scales.
* the restrictions indicate that the study is limited to the Russian sample.
We hope that we have been able to make our article better.
Best regards,
All the authors
